# Adjuvanted influenza vaccination increases pre-existing H5N1 cross-reactive antibodies

Mariana Alcocer Bonifaz[1], Disha Bhavsar [2,3], Claire-Anne Siegrist [1], Arnaud Didierlaurent [1] & Benjamin Meyer [1]✉

Highly pathogenic H5N1 avian influenza viruses of clade 2.3.4.4b cause sporadic human infections and currently raise concerns about a new influenza pandemic. Heterogeneities in disease severity have been observed in the past and are reported among infected farm workers in the United States. These may be attributed to differences in pre-existing H5N1 cross-reactive antibodies. In this study, we characterize H5N1 cross-reactive antibody landscapes in the current population (#NCT05794412 and #NCT01022905) and assess the effect of AS03-adjuvanted pandemic H1N1 and non-adjuvanted seasonal influenza vaccination on H5N1 cross-neutralizing and IgG antibody titers targeting a range of influenza virus-derived antigens. We detect H5N1 cross-neutralizing antibodies using a vesicular stomatitis virus-based pseudovirus system that correlate well with antibodies inhibiting the spread of authentic H5N1 viruses, anti-group 1 hemagglutinin stalk and anti-trimeric hemagglutinin antibodies. Additionally, we find that AS03-adjuvanted pandemic H1N1 vaccination increases H5N1 cross-reactive antibodies significantly in a pandemic H1N1 immunologically partially naïve population. Furthermore, we show that immune imprinting causes distinct H5N1 cross-reactive antibody patterns pre-vaccination.

Highly pathogenic avian influenza viruses of the subtype H5N1 have first been detected in southern China in 1996 in wild birds[1]. Since then, H5N1 viruses have evolved into multiple clades and spread worldwide through migratory birds[2–4]. Since 2020, H5N1 viruses of the clade 2.3.4.4b has led to mass mortality among wild avian species and marine mammals as well as on poultry and mink farms. In February 2024, H5N1 clade 2.3.4.4b viruses unexpectedly emerged in dairy cows in the United States, raising concerns about a H5N1 pandemic[5–10]. The ongoing outbreak among cattle gives the virus the chance to further adapt to mammalian species either through acquiring mutations directly in cattle or through reassortment during spillover event to humans infected with seasonal influenza viruses. As of 25th of June 2025, H5N1 infections among dairy cows have been detected on 1,074 dairy herds in 17 states[11].

In total, 70 confirmed human H5N1 infections have been detected in the US since February 2024 with 41 cases exposed on dairy cattle farms, 24 cases exposed on poultry farms, 2 exposed to other animals and 3 with unknown exposure[11]. Confirmed cases are characterized mainly by conjunctivitis and mild respiratory disease and human-to-human transmission has not been established[12]. Only 2 severe cases, of which one died, have been reported in North America and both have been associated with H5N1 lineage D1.1 that circulates mainly among poultry[13,14]. In addition, serosurveys revealed that 7% of farm workers working on H5N1 affected farms harbor H5N1-specific antibodies indicating that many infections remain undetected[15]. In the past, H5N1 infections in humans have been associated with a high case-fatality ratio of approximately 50%[1]. Historic seroprevalence in known H5N1 exposed persons was estimated to be 1.9% at the most, i.e. con-

[1]Center of Vaccinology, Department of Pathology and Immunology, Faculty of Medicine, University of Geneva, Geneva, Switzerland. [2]Department of Microbiology, Icahn School of Medicine at Mount Sinai, New York, NY, USA. [3]Center for Vaccine Research and Pandemic Preparedness (C-VaRPP), Icahn School of Medicine at Mount Sinai, New York, NY, USA. ✉e-mail: benjamin.meyer@unige.ch

siderably lower than in dairy farm worker in the US, indicating that asymptomatic or mild H5N1 infections were less common in the past[16,17].

Previous studies have shown that pre-existing influenza virus immunity could influence severity of disease. Influenza viruses can be separated into 2 antigenically distinct groups based on their hemagglutinin (HA) sequences, i.e. group 1 (e.g. H1N1, H2N2, H5N1) and group 2 (e.g. H3N2, H7N9)[18]. The first encounter with an influenza virus during childhood results in a skewed subsequent immune response towards this strain, a phenomenon now termed immune imprinting. Using mathematical modeling, Gostic et al. showed that immune imprinting with group 1 but not group 2 influenza viruses protected from death due to H5N1 infection[19]. Previous infection with H1N1 reduced disease severity of H5N1 infections in a ferret model[20]. It is therefore importance to assess the level of cross-reactive antibodies in the population and how these could be increased using available vaccination strategies in case of a H5N1 pandemic.

Here, we study neutralizing and binding H5N1-specific antibodies in individuals with different immune imprinting who received AS03-adjuvanted pandemic H1N1 (pH1N1/AS03) or non-adjuvanted seasonal influenza vaccines, before and after vaccination. Primary outcomes of the pH1N1/ASO3 vaccination trial have been published previously[21–23]. Our results provide critical insights into how immune imprinting leads to birth year specific differences in antibody levels and how these could be overcome using a low-dose pH1N1/AS03 influenza vaccine in a pH1N1 immunologically partially naïve population.

## Results

### Adjuvanted influenza vaccination boosts pre-existing H5N1 cross-neutralizing antibodies

In this study we aim to assess the levels of H5N1 clade 2.3.4.4b cross-neutralizing antibodies in an unexposed population and investigate the impact of pH1N1/AS03 and seasonal influenza vaccination on H5N1-specific antibody titers. We first determined H5N1 cross-neutralizing activity in pre- and post-vaccination paired serum samples from 44 healthy adults immunized with the seasonal influenza vaccine in 2023. Sera were collected between September 2023 and January 2024 at the University Hospital of Geneva (HUG), taking advantage of a prospective longitudinal cohort study aimed at following the immune response to influenza infection (2023 cohort, Table 1). Samples were collected at a median of 28 days (IQR 14.75-44.25) prior to vaccination and 68 days (IQR 53.5-83) following vaccination. Since the seasonal influenza vaccine was not administered as part of the study, it is unknown which of the licensed seasonal influenza vaccines participants have received. Exposure to H5N1 viruses in the past has not been formally assessed in this cohort, but due to the absence of human cases in Switzerland and the low total number of human H5N1 infections world-wide, the likelihood of previous exposure to H5N1 is considered negligible. Serum neutralization activity was assessed using vesicular stomatitis viruses (VSV) pseudotyped with HA and neuraminidase (NA) of a clade 2.3.4.4b influenza virus (strain: A/Pelican/Bern/1/2022) (VSV-H5N1). H5N1 pseudoneutralizing antibodies were detected in all 44 participants before seasonal influenza vaccination, with a mean log10 antibody titer of 2.01. Seasonal influenza vaccination led to a significant 1.3-fold increase in H5N1 cross-neutralizing antibodies reaching a mean log 10 antibody titer of 2.13 post vaccination, slightly higher than before vaccination (Fig. 1A). Next, we asked whether the use of an adjuvant in the influenza vaccine, could lead to a more pronounced increase in H5N1 cross-neutralizing antibody levels, given the established effect of adjuvant in enhancing the breadth of the antibody response[24–27]. For this, we used biobanked samples from a study aimed at evaluating the immune response to the monovalent pH1N1/AS03 influenza vaccine in 2009. Out of the 133 participants, 121 (91%) had H5N1 cross-neutralizing antibodies before vaccination with a mean log10 antibody titer of 2.097. pH1N1/AS03 vaccination

**Table 1 | Participant characteristics**

| Group | 2009 cohort | 2023 cohort |
|---|---|---|
| **Number of participants** | 133 | 44 |
| **Age in years: median (IQR)** | 51 (41-63) | 37 (28-46.25) |
| **Sex:** | | |
| **Females (%)** | 75 (56.4%) | 32 (72.7%) |
| **Males (%)** | 58 (43.6%) | 12 (27.3%) |
| **Ethnicity** | | |
| **Caucasian** | 114 (85.7%) | |
| **Middle-Eastern** | 3 (2.26%) | |
| **Asian** | 7 (5.26%) | |
| **North African** | 0 | |
| **Subsaharan African** | 5 (3.76%) | |
| **South American** | 3 (2.26%) | |
| **Other** | 1 (0.75%) | |
| **Birth cohort (birth year)** | | |
| ≤1954) | 51 | 0 |
| [1955-1965] | 40 | 4 |
| [1966-1976] | 27 | 7 |
| ≥1977] | 15 | 33 |
| **Seasonal flu vaccination in 2008** | | |
| yes | 61 (45.9%) | na |
| no | 70 (52.6%) | na |
| unknown | 2 (1.5%) | na |
| **Seasonal flu vaccination in 2009** | | |
| yes | 70 (52.6%) | na |
| no | 63 (47.4%) | na |
| **Seasonal flu vaccination in 2023/2024** | | |
| yes | na | 44 |
| no | na | 0 |

significantly increased H5N1 cross-neutralizing antibodies by 3.7-fold reaching a mean log10 titer of 2.66 post vaccination (Fig. 1B). For comparison, we measured H5N1 neutralizing antibodies in the 1st international H5N1 standard plasma pool (NIBSC: #07/150), which originates from individuals vaccinated with the A/Vietnam/1194/2004 (H5N1) (NIBRG-14) vaccine with our assay and detected a log10 antibody titer of 2.18, similar to what we found in our cohorts.

### HA- but not NA-specific antibodies correlate with pseudovirus neutralization titers

We next assessed the specificities of the antibodies that were responsible for H5N1 neutralization. Given the low likelihood of prior H5N1 exposure in the general population, the detection of cross-neutralizing activity against H5N1 was unexpected and prompted further investigation into the targets of these antibodies. While neutralizing antibodies primarily target the HA protein[28–31], recent studies have highlighted the important role of NA and the conserved HA stalk domain in mediating cross-protection[32–35]. Therefore, we assessed antibody binding to full-length HA and NA proteins, as well as to HA stalk domains. We performed a multiplex bead-based serological assay to determine IgG antibody levels against recombinant trimeric HAs from seasonal H1 (A/California/07/2009) and H3 (A/Perth/16/2009) and avian H5 clade 2.3.4.4b (A/Pelican/Bern/1/2022) strains, chimeric HA's from antigenic group 1 and 2 HA's (cH6/1 and cH7/3) to measure HA stalk-specific antibodies and seasonal N1 (A/California/04/2009) and N2 (A/Hong Kong/4801/2014) NAs as well as an avian N1 (A/Anhui/1/2005) NA. In both cohorts, highest correlation (Spearman r ranging between 0.57-0.75) was observed between VSV-H5N1 neutralization and IgG levels against H5 HA in pre- as well as post-vaccination sera

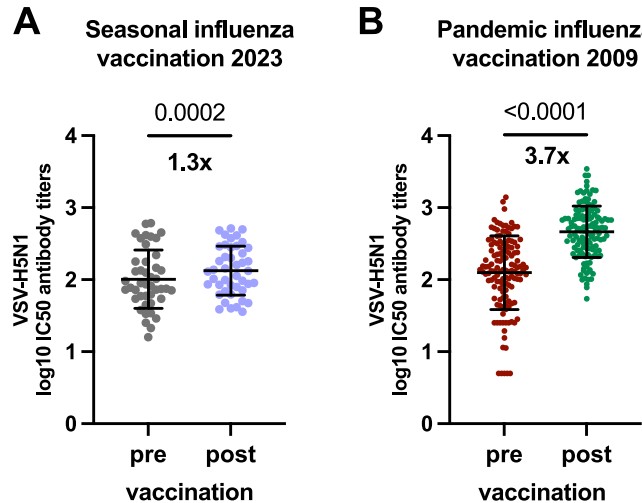

**Fig. 1 | H5N1 cross-neutralizing antibodies after adjuvanted and non-adjuvanted influenza vaccination.** H5N1 cross-neutralizing antibody titres in serum of individuals that have received a non-adjuvanted seasonal influenza vaccine (n = 44) in 2023 (**A**) or the pH1N1/AS03 vaccine (n = 133) in 2009 (**B**) determined using recombinant virus VSV*ΔG(HA_{H5},NA_{N1}) pseudotyped with H5N1 clade 2.3.4.4b HA and NA (A/Pelican/Bern/1/2022). Mean and SD is shown. Two-tailed Wilcoxon matched-pairs signed rank test was performed to compare pre and post vaccination antibody titres. Source data are provided as a Source Data file.

(Fig. 2A–D). Additionally, significant but moderate levels of correlation were found between VSV-H5N1 neutralization and group 1 stalk-specific IgG as well as trimeric H1 HA-specific IgG titers (Fig. 2A–D). No significant correlation between NA-specific IgG levels and VSV-H5N1 neutralization was observed. Absolute antibody titers for all antigens are shown in supplementary fig. 1. These findings indicate that, VSV-H5N1 neutralization is mediated by HA-specific antibodies likely directed against the more conserved stalk region.

### Authentic H5N1 virus neutralization is mediated by inhibition of virus spread

To validate our findings, we determined inhibition of virus entry in a set of samples (n = 12) from the 2009 cohort before pH1N1/AS03 vaccination using authentic HPAI H5N1 clade 2.3.4.4b virus (strain: A/Pelican/Bern/1/2022). Although antibody neutralize VSV-H5N1 and bind avian HA, no inhibition of virus entry was observed, even at low serum dilutions (down to 1:10). Antibodies, however, may also prevent viral spread from cell to cell. We therefore looked at plaque size reduction assay by adding sera after the virus had infected cells. We found that the 12 sera tested from the 2009 cohort inhibit the spread of authentic H5N1 virus in a dose dependent manner (Fig. 3A). The values obtained in this assay correlated well with the ones obtained using VSV-H5N1 (Fig. 3B, Spearman r = 0.92, p = 0.0001) indicating that VSV-H5N1 neutralizing antibody titers can serve as a good surrogate for the inhibition of authentic H5N1 virus spread.

### Recent seasonal influenza vaccination leads to lower H5N1 cross-reactive antibody titers post vaccination

Next, we investigated which factors can influence the levels of H5N1 cross-reactive antibodies. The following analyses were performed only in the 2009 cohort that has received the pH1N1/AS03 vaccine. We found no difference in VSV-H5N1 neutralizing antibodies as well as H5 trimer- and cH6/1 stalk-binding antibodies between male and female participants (Supplementary fig. 2). Since a subset of participants in the 2009 cohort had received non-adjuvanted seasonal influenza vaccines before the pH1N1/AS03 vaccination, we investigated the effect of recent seasonal influenza vaccination on H5N1 cross-reactive

antibody levels. We grouped participants based on the timing of their non-adjuvanted seasonal influenza vaccination relative to the pH1N1/AS03 vaccination: those who received it in 2008 (i.e.one year before), those who received it in 2009 (i.e. app. one month before, median 36 days, IQR 31.25-48) and those who received it both in 2008 and 2009. Individuals who had received a seasonal influenza vaccine in the weeks preceding vaccination with pH1N1/AS03 had significantly higher H5N1 cross-neutralizing and H5 trimer specific cross-reactive IgG antibody titers compared to those who had received no seasonal vaccine (Fig. 4A and B). A similar trend was observed for group 1 stalk-specific IgG antibodies, but the effect was not significant (Fig. 4C). After vaccination with pH1N1/AS03 vaccine, H5N1 cross-neutralizing and cross-reactive IgG antibody titers became lower in individuals that had recently received a seasonal influenza vaccine compared to those who had not. This indicates that seasonal influenza vaccination shortly before pH1N1/AS03 vaccination blunted the H5N1 cross-reactive antibody response (Fig. 4A–C). For subjects that only received a seasonal influenza vaccine in the previous year, we did not observe a similar effect, although the number of participants in this group is low (Fig. 4A–C).

### Birth year specific differences in H5N1 cross-reactive antibody levels

Since immune imprinting has been shown to influence mortality of H5N1 infections[19], we asked whether H5N1-specific antibody levels differ between individuals imprinted with HA antigenic group 1 or group 2 influenza viruses. This question was investigated only in the 2009 cohort since only few individuals of the 2023 cohort had been born before 1977. During the 20th century, influenza virus circulation followed a distinct pattern, with H1N1 and H2N2 (group 1) exclusively circulating between 1918-1956 and 1957-1967, respectively, and H3N2 (group 2) exclusively circulating between 1968-1976 (Supplementary Fig. 3A). After 1977, H1N1 and H3N2 influenza viruses cocirculated, but H3N2 dominated most years. We calculated imprinting probabilities of individuals in Switzerland based on virus circulation using the R package from Gostic et al. (Supplementary Fig. 3B). Groups, based on the year of birth, were defined considering the calculated imprinting probabilities. Individuals born in 1955 and 1956 had a higher probability of being imprinted with H2N2 (51.7% and 71.8%, respectively) than with H1N1 and were therefore grouped with individuals born between 1957 and 1965, whereas individuals born 1966 and 1967 had a higher probability to be imprinted with H3N2 than with H2N2. Subjects born in 1977 and after were grouped together, irrespective of imprinting probabilities. At baseline, prior to pH1N1/AS03 vaccination, individuals born in and before 1954 (mainly imprinted with H1N1) showed the highest VSV-H5N1 neutralization activity and group 1 HA IgG antibody titers (Fig. 5 middle panel). Compared to group 1 HA imprinted individuals, the significant differences were observed for individuals mainly imprinted with H3N2, i.e. birth year groups 1966-1976 and those born after 1977. Interestingly, administration of pH1N1/AS03 vaccine, abolished the differences in H5N1 cross-reactive antibody levels across birth year groups (Fig. 5 right panel). This largely reflects a significant stronger increase in antibody levels in individuals imprinted with group 2 HA viruses (Supplementary Fig. 4A–D). To assess whether the observed effect was influenced by receiving a seasonal influenza vaccination one month prior to pH1N1/AS03 vaccination, we performed the same analysis only in individuals who had not received a seasonal influenza vaccine in 2009. Similar antibody patterns between birth year cohorts were found (Supplementary Fig. 5A–C), indicating that the observed effects are caused by immune imprinting and not by prior seasonal influenza vaccination. Our findings indicate that higher functional and binding H5N1 specific antibody levels have been found in individuals imprinted with HA antigenic group 1 viruses (H1N1 and H2N2), but imprinting patterns can be overcome by vaccination.

**A**
## 2023 cohort pre vaccination

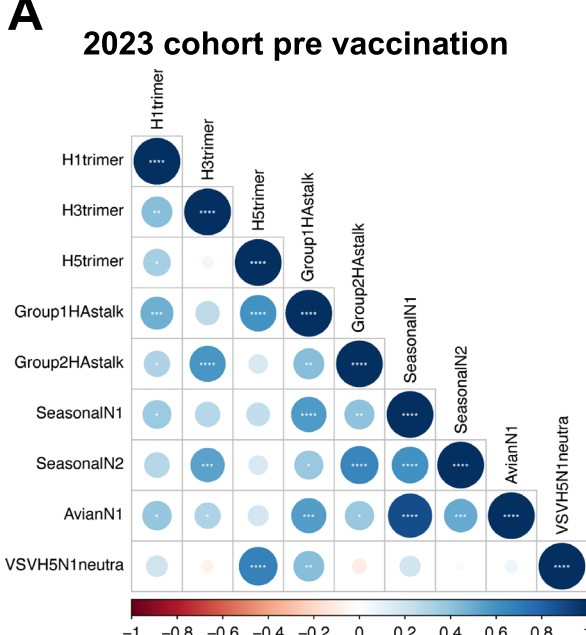

**B**
## 2023 cohort post vaccination

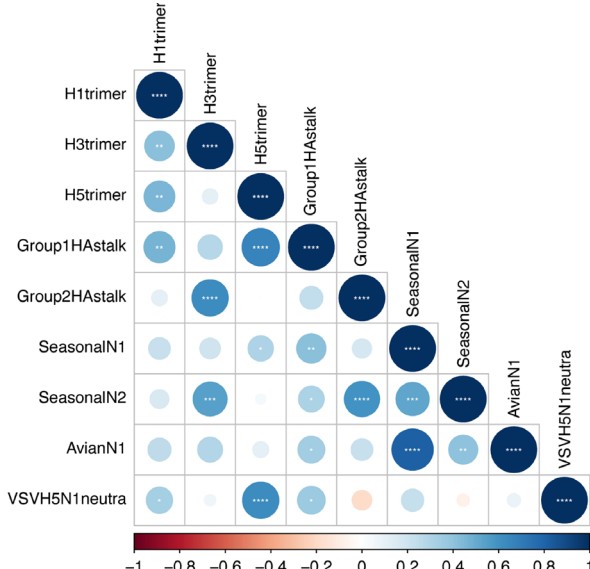

**C**
## 2009 cohort pre vaccination

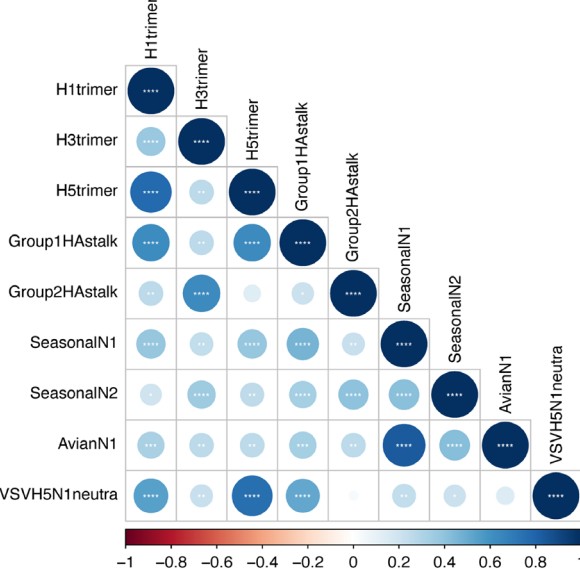

**D**
## 2009 cohort post vaccination

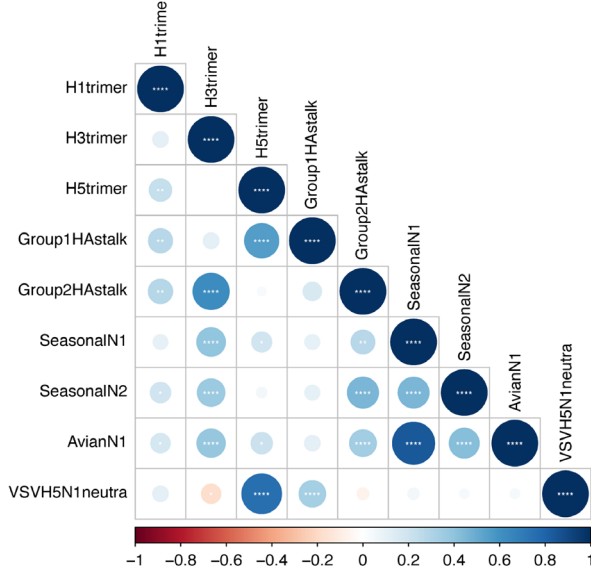

**Fig. 2 | Correlation of H5N1 cross-neutralizing and influenza virus antigen specific IgG antibody titres.** Spearman correlation analyses were performed pre and post seasonal influenza vaccination in 2023 (**A** and **B**) (n = 44) and pH1N1/AS03 vaccination in 2009 (**C** and **D**) (n = 133), respectively. Antibody titres to influenza recombinant proteins were determined by Luminex Immunoassay. Spearman correlations (two-tailed) were performed using R and plotted using corrplot package. The color scale (indicated at the bottom of each plot) and the size of the circle correspond to Spearman's rank correlation coefficient value (rs). The number of stars indicate the *p*-values (* p < 0.05; ** p < 0.01; *** p < 0.001; **** p < 0.0001). Source data are provided as a Source Data file.

## Discussion

Avian H5N1 influenza viruses of the clade 2.3.4.4b are currently circulating among poultry and dairy cows raising concerns about a potential new pandemic. Sporadic human cases have been reported and show a wide spectrum of disease severity ranging from asymptomatic to severe infections, including deaths[13,14]. In this study, we investigated how existing H5N1 cross-reactive antibodies in the population are influenced by seasonal or adjuvanted influenza vaccination and explored the influence of immune imprinting on H5N1 cross-reactive antibodies.

Using VSV pseudotyped with HA and NA of a clade 2.3.4.4b H5N1 virus, we could show that H5N1 cross-neutralizing antibodies exist in the population. While several studies have reported isolation of H5N1 cross-neutralizing monoclonal antibodies from human donors[36–38], the detection of H5N1 cross-neutralization activity in human blood samples, such as serum or plasma, varies considerably. In line with our findings, a recent pre-print found low levels of H5N1 cross-neutralizing antibodies in all 66 individuals in a cohort from Germany using a lentivirus-based pseudovirus neutralization assay[39]. In contrast, other

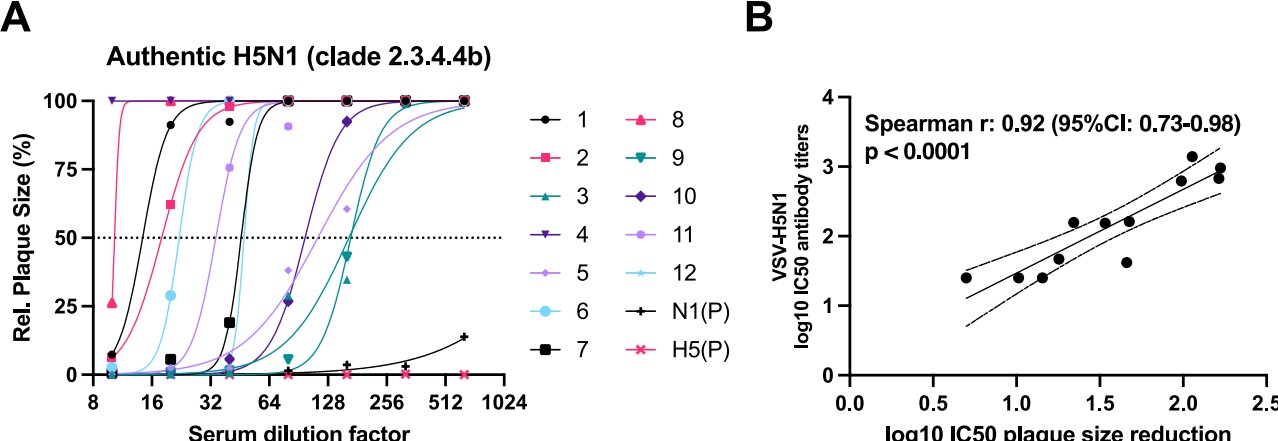

**Fig. 3 | Neutralization of authentic HPAIV H5N1. A** Serum neutralizing antibodies measured by authentic highly pathogenic H5N1 virus clade 2.3.4.4b (A/Pelican/Bern/1/2022) using a plaque size reduction assay (n = 12). Chicken sera which were directed against N1 neuraminidase and H5 hemagglutinin from H5N1 virus clade 2.3.4.4b (A/Pelican/Bern/1/2022) were used as positive controls (depicted as N1 (P) and H5 (P). Nonlinear fit regression curves were performed using GraphPad Prism following Sigmoidal, 4PL, X is concentration and setting top=100 and bottom=0

constrains. **B** Spearman correlation (two-tailed) between plaque count reduction neutralization titres obtained from neutralization assay using VSV-H5N1 pseudo-viruses and plaque size reduction neutralization titres obtained from neutralization assay using authentic H5N1 viruses. Error bands (dashed lines) represent 95% confidence bands of the best-fit line (solid black line). Source data are provided as a Source Data file.

studies found detectable H5N1 neutralizing antibody titers only in a small subset of participants, using both authentic and pseudovirus neutralization[40–43], or could not detect any H5N1 specific neutralizing activity at all[44,45]. These observations can potentially be attributed to differences in the sensitivity of neutralization assays using pseudo-viruses or authentic H5N1 as previously reported for influenza viruses[42] and SARS-CoV-2[46]. Interestingly, pseudotype-based neutralization assays have been found to detect broadly reactive stalk-specific anti-bodies more efficiently than assays using authentic influenza viruses, indicating that they are better suited to find low-levels of H5N1 cross-neutralizing antibodies[47,48]. Wang et al. showed that for Newcastle disease virus pseudo virus-based neutralization assay successfully detected neutralizing antibodies in samples where authentic neu-tralization assay reported negative results. A possible explanation as to why pseudovirus neutralization assays are more sensitive compared to assays using authentic viruses are morphological differences between the VSV and viral particles of authentic viruses[49–51]. Moreover, lower limits of detection are assay dependent and can vary depending on virus species, the titer of the virus or the volume of serum sample used. It is thus important to report results relative to a reference material to enable comparisons between data produced by different methods[52].

In our study, we did not detect antibodies inhibiting entry of authentic H5N1 viruses, similar to what has been found by Daniel et al.[39]. We could, however, show there was an inhibition of the spread of authentic H5N1 viruses in a dose dependent manner using a plaque size reduction assay, which correlated strongly with VSV-H5N1 pseudovirus neutralization titers. Despite the different mechanisms in neutraliza-tion, VSV-based pseudovirus neutralization titers predict antibody titers that inhibit authentic H5N1 virus spread, indicating that VSV-H5N1 viruses can be used as a surrogate system. Similarly, strong correlation between authentic and pseudovirus based neutralization assays have been found in other studies[40].

This is the first study that found that pH1N1/AS03 vaccination significantly increased H5N1 cross-neutralizing antibodies (3.7-fold) in a population immunologically partially naïve to pH1N1. In addition, we found only a marginal increase in H5N1 cross-neutralization (1.3-fold) after seasonal influenza vaccine, in line with previous studies that show low[40] or no increase[44] but in a pH1N1 exposed population. Differences in antigen composition between adjuvanted and seasonal influenza

vaccine as well as in pre-existing immunity in the 2009 and 2023 cohorts prevent a direct comparison between the two vaccines. Moreover, due to the absence of an appropriate control cohort we cannot determine whether the increase in H5N1 cross-neutralizing antibodies in the 2009 cohort is the result of the presence of AS03 in the vaccine or vaccinating a pH1N1 immunologically partially naïve population. In 2009, the population was largely immunologically naïve to the pandemic H1 HA head but had cross-reactive memory B cells (MBC) against the more conserved HA stalk[53]. In the absence of HA head specific MBCs that normally outcompete stalk HA specific MBCs presumably due to better accessibility of the HA head[54], pH1N1/AS03 vaccination led to a predominant increase in group 1 HA stalk-specific antibodies[55], which is in line with our findings. Due to the presence of pre-existing head-specific MBCs, seasonal influenza vaccination results in a predominant HA head specific immune response with minimal induction of HA stalk specific antibodies[54] as it was the case in the 2023 cohort. It seems likely that the increase in group 1 HA stalk specific antibodies is largely responsible for the increase in H5N1 cross-neutralization capacity in the 2009 cohort. However, data from other studies suggests that AS03 may also contribute to the increase of H5N1 cross-neutralizing antibodies. Responses to AS03 adjuvanted and non-adjuvanted vaccination have been compared using H5 HA as an anti-gen in H5N1 unexposed individuals. These subjects likely have cross-reactive HA stalk-specific MBCs but no H5 head specific MBCs, which resembles the situation in 2009 for pH1N1. Group 1 HA stalk specific antibodies were induced by both AS03 adjuvanted and non-adjuvanted H5N1 vaccination, whereas H5 head specific antibodies were only generated after AS03 vaccination indicating that AS03 is important for inducing naïve B cells[56]. Similarly, pH1N1/AS03 vaccina-tion also induced naïve B cell activation[57] leading to the generation of HA head specific antibodies. Interestingly, our data shows that trimeric H5 HA antibody titers correlate better with H5N1 cross-neutralizing antibody titers than group 1 stalk HA antibodies across all cohorts and time points (Fig. 2), indicating that also antibodies targeting conserved sites at the HA head might contribute to H5N1 cross-neutralization. Interestingly, HA head specific monoclonal antibodies that are broadly neutralizing across H1, H5 and H3 influenza viruses, e.g., F045-092, have previously been found in humans[37]. However, the contribution of HA head-specific antibodies to H5N1 cross-neutralization in the 2009 cohort needs to be further investigated.

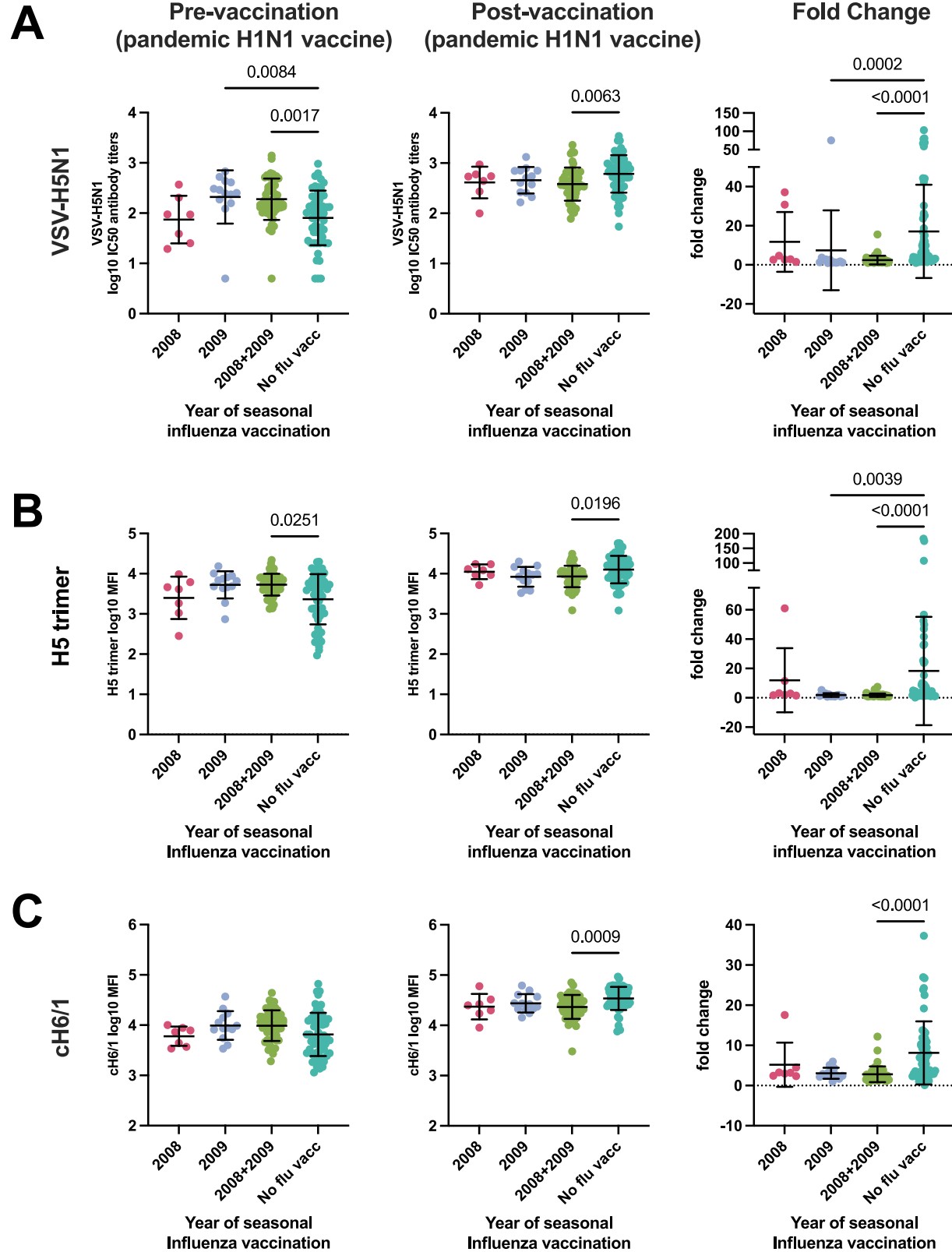

**Fig. 4 | The effect of previous seasonal influenza vaccination on VSV-H5N1 cross-neutralizing antibody titres, H5- and group 1 HA stalk- reactive antibody titres.** Log10 transformed antibody titres pre and post pH1N1/AS03 vaccination and fold changes grouped by whether participants have received a seasonal influenza vaccine only in 2008 (n = 7), 2009 (n = 13), in both years (2008 + 2009) (n = 50) or not (n = 56). **A** Neutralizing antibodies against VSV-H5N1 pseudoviruses. **B** IgG antibody binding titres to H5 HA from HPAIV H5N1 (A/Pelican/Bern/1/2022). **C** Antibody binding titres to group 1 HA stalk (cH6/1). All antibody titres have been log10 transformed. Mean and SD is shown. Kruskal-Wallis test (two-tailed) with Dunn's test were performed to compare the mean rank of each column and to correct for multiple comparisons. Source data are provided as a Source Data file.

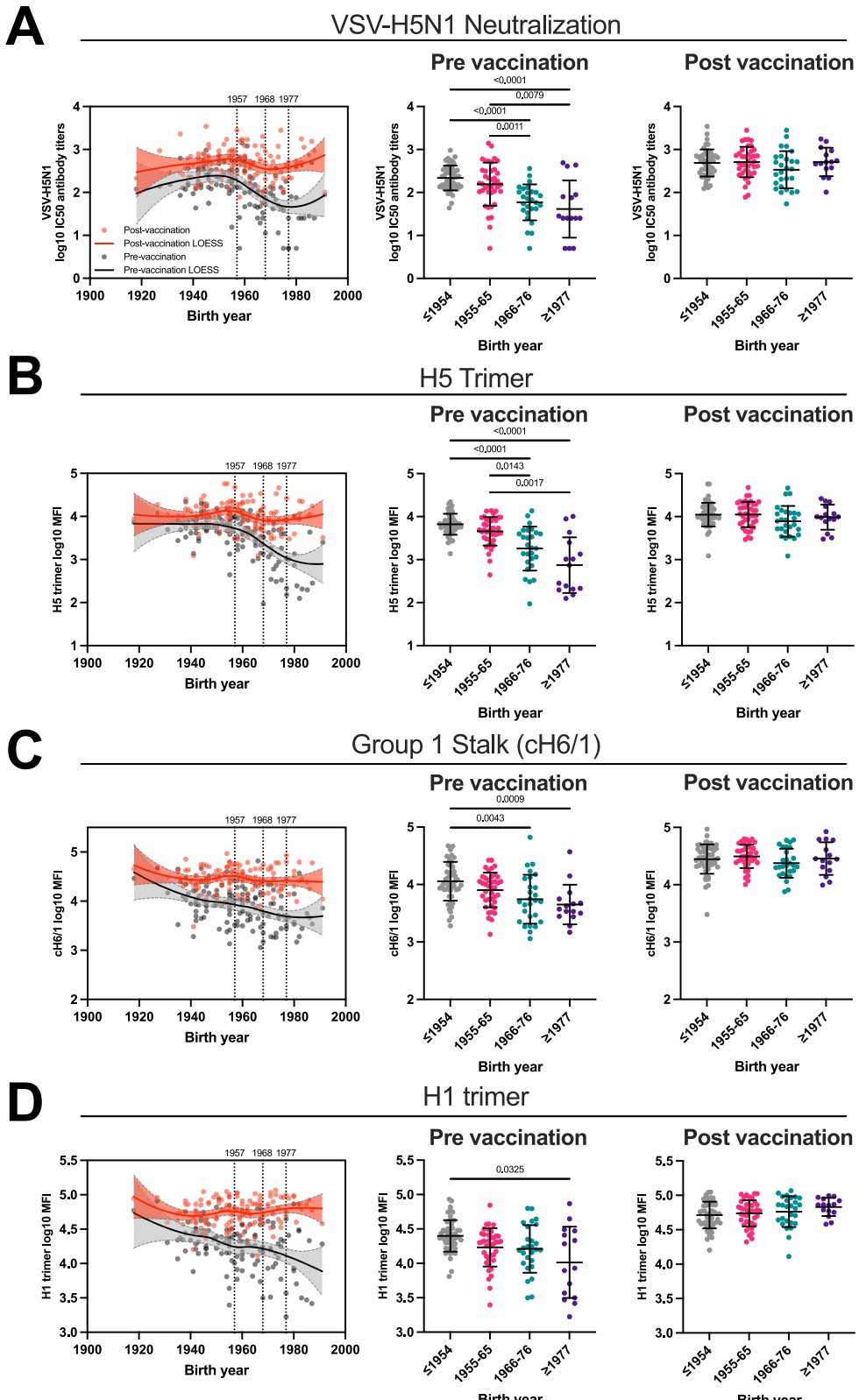

**Fig. 5 | Effect of immune imprinting on H5N1 cross-reactive antibody responses pre and post pH1N1/AS03 vaccination. A** VSV-H5N1 pseudovirus (A/Pelican/Bern/01/2022) cross-neutralizing antibody titres, (**B**) trimeric H5 HA, (**C**) group 1 stalk cH6/1 and (**D**) trimeric H1 HA log10-transformed IgG antibody titres. Left panels: the solid lines represent the fitted curve calculated using the LOESS method (n = 133). Black lines indicate pre-vaccination titres, whereas red lines indicate post-vaccination titres plotted against birth years of individuals. Dashed lines represent 95% confidence bands of the best-fit line. Middle and right panels: birth year cohorts are defined by imprinting probabilities. ≤1954 imprinted most likely with H1N1 (n = 51), 1955-65 imprinted most likely with H2N2 (n = 40), 1966-76 imprinted most likely with H3N2 (n = 27), ≥1977 imprinted either with H1N1 or H3N2 (n = 15). Mean and SD is shown. Kruskal-Wallis test (two-tailed) with Dunn's test were performed to compare the mean rank of each column and to correct for multiple comparisons. Source data are provided as a Source Data file.

We observed reduced H5N1 clade 2.3.4.4b cross-neutralizing antibody titres in participants who had recently (median 36 days) received a seasonal influenza vaccine. Similarly, Roy-Ghanta et al.[57] observed that participants who previously received trivalent seasonal influenza vaccine (TIV) had a diminished pandemic H1N1-specific humoral response following AS03-adjuvanted H1N1 vaccination. Other studies have found a blunted vaccine response in individuals that have repeatedly been vaccinated in consecutive seasons[58]. This blunting of H5N1 responses elicited by H1N1-AS03 vaccine was not observed in participants who had received their most recent seasonal influenza vaccine a year earlier, although this may be attributed to a smaller sample size. The potential mechanisms behind this phenomenon remain unclear. It has been suggested that pre-existing antibodies inhibit the proliferation of influenza antigen-specific B cells[58,59]. Another explanation could be the generation of immune complexes between pre-existing antibodies and vaccine antigens that lead to a reduction of available antigen and subsequently a diminished B cell response[58,60]. Moreover, others postulated that repeated immunization could result in the refocusing of antibody responses towards more conserved antigens[61]. Overall, the high sequence conservation between group 1 influenza viruses (H1N1 and H5N1) as well as the continuous seasonal influenza exposure could explain the presence of pre-existing antibodies that influence the antibody titres and breadth.

Previous influenza pandemics have affected age groups differently[62], and the year of birth of H5N1 infected patients has been shown to strongly influence disease outcome[19]. We found that group 1 HA imprinted individuals had higher H5N1 cross-reactive antibodies compared to group 2 HA imprinted individuals. Similar results have been found in another recent study[63], confirming that similar imprinting patterns on H5N1 cross-reactive antibody levels exist across two independent cohorts from the Northern Hemisphere, where individuals are likely to share comparable influenza or vaccine exposure histories. These findings could explain the decreased susceptibility and lower mortality to H5N1 infection observed in older adults[19]. Furthermore, the absence of hemagglutination inhibition (HAI) antibody titres against avian H5N1[44,64,65], combined with the correlation between group 1 HA-reactive antibody levels and H5N1 cross-neutralizing antibody titres, suggests that antibodies targeting the conserved HA stalk domain may play a key role in H5N1 cross-neutralization. Importantly, pH1N1/AS03 vaccination was able to overcome immune imprinting patterns of H5N1 cross-reactive antibody titres and induce higher increases of antibodies in group 2 HA imprinted individuals in a pH1N1 immunologically partially naïve population. In a recent study, participants that likely harbor cross-reactive group 1 HA stalk-specific MBCs but no H5 HA head specific MBCs, a situation similar to the 2009 cohort, received a non-adjuvanted H5N1 vaccines at a high dose of 45ug of HA (12x higher than the HA content of the pH1N1/AS03 vaccine). Similar to our findings with AS03 adjuvanted pH1N1 vaccination, non-adjuvanted high dose vaccination resulted in higher fold changes in group 2 HA compared to group1 HA imprinted individuals[63], indicating that the pre-existing immunity and not the adjuvant is largely responsible for the higher increase of HA antibodies in group 2 imprinted participants. However, the differences in antibody levels between birth cohorts were only reduced but did not vanish completely despite the use of a 12-fold higher HA dose[63]. These results suggest that adjuvants could help to completely overcome suboptimal vaccine responses in certain age groups using a low-dose vaccines which is critical during limited global pandemic influenza antigen manufacturing capacity.

Our study has several limitations. First, we were not able to compare adjuvanted and non-adjuvanted influenza vaccination with the same antigen. Second, adjuvanted influenza vaccination was evaluated only in 2009, in individuals that have been immunologically naïve to the pandemic H1N1 influenza virus strain, but not in the current population that has a different immunological background. Therefore, we cannot demonstrate which effect AS03 adjuvanted vaccination would have on the current pH1N1 immune population. Third, we lack samples from older age groups in the 2023 cohort, preventing us from studying the effect of immune imprinting in a contemporary cohort. Last, we did not investigate whether the effect of AS03 adjuvantation on immune imprinting is temporary or long-lasting and if non-adjuvanted seasonal influenza vaccination could also overcome immune imprinting.

In conclusion, we could show that low levels of H5N1 cross-neutralizing antibodies exist in the population in 2009 and more recently in 2023. Low dose AS03-adjuvanted pandemic H1N1 vaccination was able to substantially induce H5N1 cross-reactive antibodies and could overcome the effect of immune imprinting on H5N1 cross-reactive antibody patterns in a pH1N1 immunologically partially naïve population.

## Methods

### Ethics statement

Serum samples used in this study were selected from two clinicals studies previously registered on www.clinicaltrials.gov (#NCT05794412 and #NCT01022905). Written informed consent was obtained from all participants. The studies were approved by the cantonal ethics commission at the University Hospitals of Geneva (CCER) prior to enrollment. The trials were conducted in accordance with the principles of the Declaration of Helsinki, standards of Good Clinical Practice, and Swiss regulatory requirements.

### Participants characteristics

In the 2023 cohort (#NCT05794412), serum samples ($n = 44$) were collected between September 2023 and January 2024. Selected participants self-reported non-adjuvanted seasonal influenza vaccination (1 dose) during the 2023/2024 influenza season administered outside of the study. Blood samples were collected app. one month before vaccination (median 28 days, IQR 14.75–44.25) and two months after vaccination (median 68 days, IQR 53.5–83). For the 2009 cohort (#NCT01022905), serum samples ($n = 133$) were collected between November and December 2009. All participants received one dose of pH1N1/AS03 influenza subunit/split virus vaccine (Pandemrix®, GlaxoSmithKline). Each dose contained H1 HA antigen (3.75 ug), squalene (10.69 mg), DL-α-tocopherol (11.86 mg), polysorbate 80 (4.86 mg). Blood was collected immediately prior to vaccination and one month after vaccination (median 29 days, IQR 28–36). Gender was not recorded in either study. Sex was self-reported and was not verified. Primary outcome of the 2009 cohort has been previously published[21–23]. We performed exploratory analysis by assessing immune responses against other influenza virus strains without any pre-defined outcomes. Sera were prepared and stored at -20 °C until analysis. Participants characteristics for both cohorts are summarized in Table 1.

WHO international standard for antibody to influenza H5N1 virus consisting of a pooled plasma from human recipients of A/Vietnam/1194/2004 clade 1 (H5N1) (NIBRG-14) (NIBSC code: 07/150) vaccine was obtained from NIBSC.

### Cell lines

Madin-Darby canine kidney cells (MDCK) were cultured in minimum essential medium (MEM) supplemented with Earle's salts, L-Glutamine (Thermofisher; Cat.No.31095029) and 5% fetal bovine serum (FBS) (Thermofisher; Cat. No. A5256701; Lot. No. 2575625). Cell lines were grown at 37 °C and 5% $CO_2$. For passaging, the cells were washed with PBS once, incubated 10 min with 0.05% trypsin-EDTA (Thermofisher; Cat.No. 25300054) and split in 1/10 ratio.

## Viruses

Recombinant virus VSV*ΔG(HA$_{H5}$,NA$_{N1}$) containing the HA and NA genes of highly pathogenic avian virus strain A/Pelican/Bern/1/2022 (accession No. EPI_ISL_19371737) was kindly provided by Prof. Gert Zimmer. We took advantage of the expression of avian H5 and N1 on the surface of recombinant VSV*ΔG(HA$_{H5}$,NA$_{N1}$) viruses and green fluorescent protein (GFP) for the quantification of H5N1 cross-neutralizing antibodies.

## Virus propagation

MDCK cells were seeded in T150 cell culture flasks to reach 70-80% confluency. Cell monolayer was washed twice with PBS. Parental virus stock was diluted 1/100 in 6 ml MEM media and added to the cell monolayer. Infection was allowed to proceed for 1 h at 37 °C and 5% $CO_2$. Then, 19 ml MEM media was added to the flask and cells were incubated at 37 °C, 5% $CO_2$ until cytopathogenic effect was observed. Supernatant was collected, centrifuged at 1000 g for 10 min at 4 °C and stored at -80 °C. VSV*ΔG(HA$_{H5}$,NA$_{N1}$) virus stock was stored with an additional 10% FBS.

## Virus titration

Briefly, in a 96-microplate (Brunschwig; Cat.No. 3603-48ea Corning) 75,000 MDCK cells were plated in 100ul MEM media (without FBS) per well. Cells were incubated O/N at 37% and 5% $CO_2$. 3-fold serial dilutions of parental VSV*ΔG(HA$_{H5}$, NA$_{N1}$) virus stock were prepared to infect MDCK cell monolayer for 90 min at 37 °C (starting dilution 1:30). Then, immobilizing media (MEM supplemented with 1% methylcellulose (Sigma; M0512-00G), 2% FBS and 1% Penicillin/ Streptomycin (Gibco; 15140-122) was added and incubated O/N at 37 °C, 5% $CO_2$. Before plate reading at Cytation 5 BioTek, the plate was washed once and fixed with 6% paraformaldehyde. Four images were taken per well at Cytation 5 BioTek. Plaque count was calculated by counting number of plaques per well. Virus titer (PFU/ml) was calculated by multiplying the average number of plaques X dilution factor X 20.

## VSV*ΔG(HA$_{H5}$,NA$_{N1}$) virus neutralization assay

Pseudoneutralization assay protocol was kindly provided by Prof. Gert Zimmer[66]. Briefly, in a 96-microplate (Brunschwig; Cat.No. 3603-48ea Corning) 75,000 MDCK cells were plated per well in 100ul MEM media (without FBS). Cells were incubated O/N at 37% and 5% $CO_2$ until cell monolayer was formed. Human sera were heat-inactivated at 56 °C for 45 min and serially diluted 2-fold (starting at 1:10) in a U-bottom 96-well microtiter plate. Each sample was run in duplicate. Parental VSV*ΔG(HA$_{H5}$, NA$_{N1}$) virus stock was diluted to a final concentration of 2 PFU/ul. Then, 100 ul were added to each serum dilution. No sera were added to the last row of each plate to measure the maximum plaque count. Neutralization was allowed to proceed for 30 min at 37 °C. Cell supernatant was aspirated and replaced with corresponding virus and serum mix. Infection was allowed to proceed for 90 min at 37 °C. Then, 100ul immobilizing media (MEM + 1% Methylcellulose + 1% Penicillin/Streptomycin + 2% FBS) was added per well. The plate was further incubated O/N at 37 °C and 5% $CO_2$. Before plate reading at Cytation 5 BioTek, the plate was washed once and fixed with 6% paraformaldehyde. Four images were taken per well at Cytation 5 BioTek. Plaque count was calculated by counting number of plaques per well. Maximum plaque count was the average of virus only wells. The relative plaque count reduction (%) was calculated by dividing the average plaque count per sera dilution by the maximum plaque count and multiplied by 100. The relative plaque count reduction (%) was then plotted against the different serum dilutions in GraphPad Prism to calculate the serum dilution that reduces the plaque count (PRNT50) by 50% using Nonlinear Regression Sigmoidal, 4PL, X is concentration and 0 and 100 as minimum and maximum constrains.

## Authentic HPAIV H5N1 neutralization

Plaque size reduction assay was performed using HPAIV H5N1 clade 2.3.4.4b (strain: A/Pelican/Bern/1/2022) in the Institute of Virology and Immunology (IVI) in Bern. Briefly, MDCK cells were infected with 100 TCID50 for 40 min, washed and then incubated overnight with different 2-fold serum dilutions (starting dilution: 1/10). Infected cells incubated without immune sera served as a reference and were set to 100% foci size. Two chicken sera directed against A/Pelican/Bern/1/ 2022 NA and HA were included as positive controls. Next day, the cells were fixed with formalin and stained for viral nucleoprotein. The size of the infectious foci was determined under the fluorescence microscope. Twenty photographs were taken per serum dilution, and the foci area was determined using Zeiss microscope analysis software. Relative plaque size reduction was calculated by dividing the average plaque size per sera dilution by the maximum plaque size in virus only wells and multiplied by 100. We used GraphPad Prism to calculate the serum dilution that reduces by 50% the plaque size (Relative plaque size %) using Nonlinear Regression Sigmoidal, 4PL, X is concentration and 0 and 100 as minimum and maximum constrains.

## Recombinant proteins

Influenza recombinant trimeric H1, H3, and H5 HA proteins were designed and expressed in collaboration with the Protein Production and Structure Core Facility at the École Polytechnique Fédérale de Lausanne (EPFL). Protein expression vectors were constructed using a previously published approach[67]. To obtain trimerized secreted HA proteins, the transmembrane and intracellular domain of HA was removed and a T4-foldon domain was included at the C-terminal end of the HA sequence[68]. Sequences were codon-optimized for expression in mammalian cells and an AVI tag, a twin strep-tag and 6xhis-tag were included for purification of recombinant HA from cell culture supernatant. Protein expression vectors (pTwist+CMV+BetaGlobin +WPRE+Neo) were ordered from Twist Biosciences and protein expression was performed at the EPFL core facility by 7-day transient transfection of ExpiCHO cells in ProCHO5 + 2% DMSO. The filtered supernatant was loaded on a StrepTrap XT column (GE Healthcare Ref: 29-4013-22) and eluted with 50 mM Biotin in 150 mM NaCl + 75 mM Hepes. Dialysis was performed O/N in PBS. Afterwards, the recombinant protein was concentrated with 50 kDa cut-off Amicon tube and flash frozen. Chimeric HAs (cH6/1 and cH7/3) containing a H6 or H7 HA head and an antigenic group 1 or 2 stalk domain to assess stalk reactive antibodies were kindly provided by Prof. Florian Krammer. Recombinant chimeric HA proteins were produced using baculovirus-insect cell expression system as previously described in detail[69]. Briefly, the protein constructs were designed for cH6/1 HA (H6 head domain from A/mallard/Sweden/81/02 (H6N1), H1 stalk ectodomain from A/California/04/09 (pandemic H1N1)) or for cH7/3 HA (H7 head domain from A/mallard/Alberta/24/01 (H7N3), H3 stalk ectodomain from A/Hong Kong/4801/2014 (H3N2)) with an N-terminal signal peptide, a C-terminal thrombin cleavage site, a T4 trimerization domain and a hexahistidine tag. The baculoviruses were propagated in *Spodoptera frugiperda* (Sf9) cells and the recombinant proteins were expressed in High Five cells. The recombinant proteins were purified from High Five cell supernatant at 72 h post-infection using Nickel-Nitrilotriacetic Acid (Ni-NTA) resin (Qiagen Ref: 30210) via affinity chromatography. Seasonal N1, N2 and avian N1 NA tetrameric proteins were commercially available and purchased from Sinobiological. All information relative to the recombinant influenza HA and NA proteins are summarized in supplementary Table 1.

## Protein-bead coupling for Luminex immunoassay

For coupling 1 million beads (Diasorin), 100 ul of uncoupled MagPlex microspheres bead stock ($1.25 \times 10^7$ beads/ml) were washed once with 200ul UltraPure Distilled water DNAse/RNAse free (Ref# 10977-035; Invitrogen). Beads were resuspended in 80ul activation buffer

(100 mM sodium phosphate monobasic, pH 6.2), 10ul Pierce Sulfo-NHS (N-Hydroxysulfosuccinimide; ThermoFisher Scientific: #A39269) (50 mg/ml) and 10 ul EDC (50 mg/ml) (1-Ethyl-3-[3-dimethylamino-propyl] carbodiimide-HCl; ThermoFisher Scientific: #22980). Bead were activated for 25 min at RT in the dark with end-over-end rotation and washed three times with 500 ul coupling buffer (50 mM 3-(N-morpholino) ethanesulfonic acid hydrate (MES) pH 5.0). Afterwards, beads were resuspended in a final volume of 125 ul coupling buffer containing 6.25 ug of protein of interest and incubated for 2 h at RT in the dark in an end-over-end rotator. Coupled beads were washed once with 500 ul PBS-TBN (PBS containing 0.05% Tween 20, 1% bovine serum albumin, 0.1% sodium azide), incubated in PBS-TBN for 1 h at RT on an end-over-end rotator in the dark, then washed twice with PBS-TBN. After vortexing and centrifugation, beads were resuspended in 100ul PBS-TBN and counted using a counting slide. Beads were stored at 4 °C in the dark for 6–8 weeks.

### Luminex immunoassay
Coupled bead stock was vortexed for 1 min (Ref.655096; Thermo Scientific) 1000 beads were added to each well. Supernatant was removed and sera was diluted (1:1000, in blocking buffer: PBS containing 1% BSA, 0.05% Tween20) and added (50 ul/well). All samples were run in duplicate. Plates were sealed with aluminum foil and incubated for 2 h at RT and 800 rpm. Plates were washed three times with 200 ul washing buffer (PBS containing 0.1% BSA, 0.05% Tween20). Secondary antibody PE-conjugated mouse anti-human IgG (Cat# NB110-8347PE; Novus Bio) was diluted (1:1000) in blocking buffer and 50ul per well was added. Plates were incubated for 1 h at RT and 800 rpm in the dark. Plates were washed three times with 200 ul washing buffer. Beads were resuspended in 100 ul washing buffer and agitated for 15 min at 800 rpm before plates were read on a Magpix instrument. The background median fluorescent intensity (MFI) from wells without serum was subtracted from each sample MFI and raw values were log10 transformed. Standards were included on each plate to ensure comparability. GraphPad Prism software was used to create the graphs and perform statistical analysis.

### Statistics and Reproducibility
All data was collected using Excel (Office 365) and all analysis was performed using GraphPad Prism Version 10.5.0 and R Statistical Software version 4.5.1. No statistical method was used to predetermine sample size. No data were excluded from the analyses. The experiments were not randomized. The Investigators were not blinded to allocation during experiments and outcome assessment. All samples were measured in duplicates. All antibody data was log10-transformed and analyzed using statistical tests indicated in the figure legends. P-values are given as numbers except for correlation matrices where they are indicated using asterisks for graphical reasons as follows: * $p < 0.05$; ** $p < 0.01$; *** $p < 0.001$; **** $p < 0.0001$.

### Reporting summary
Further information on research design is available in the Nature Portfolio Reporting Summary linked to this article.

## Code availability
Code is accessible on https://yareta.unige.ch under https://doi.org/10.26037/yareta:huvlkrhbyvdmdbw64iyb7ij7va.

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

## Acknowledgements

We sincerely thank Prof. Florian Krammer for providing stalk HA antigens and Prof. Gert Zimmer for sharing VSV-H5N1 and protocols for the pseudoneutralization assay. We also thank Lisa Butticaz and Thomas Vivet for their excellent technical help. We thank Florence Pojer and Kelvin Lau from the EPFL Protein Production Facility for helping with trimerized HA antigen expression. We thank Yves Alexandre Cambet, Vincent Jaquet and Adriana Renzoni for help with the fluorescent microscope and the Luminex reader. We also thank Olha Puhach and the study teams for sample collection. Last, we thank Kenz Le Gouge for help with data analysis. This project was supported by the Swiss National Science Foundation Ambizione program (grant number: 193475) and a research grant from Moderna Inc. We are grateful for the participants who were willing to donate their samples and agree to participate in our research. The funders had no role in the study design, data collection and analysis, decision to publish, or preparation of the manuscript.

## Author contributions

M.A.B. and B.M. conceptualized the study, M.A.B. performed the experiments and analyzed the data. D.B. produced the stalk HA antigens, C.A.S. collected the clinical samples, M.A.B. and B.M. analyzed and interpreted the data, A.D. and B.M. supervised the work, All authors read and approved the final version of the manuscript. M.A.B. and B.M. verified the underlying data.

## Competing interests

B.M. has received research support from Moderna and consulting fees from Rocketvax AG. A.D. is a member of scientific advisory boards for Bioaster and Sanofi, is a consultant for Boost Biopharma, Botanical solutions and Vaccine Formulation Institute, and has research collaborations with Moderna, GSK and Sanofi. The remaining authors declare that they have no competing interests.
