## [Transparent Peer Review file · Nature Communications]

Adjuvanted influenza vaccination increases pre-existing H5N1 cross-reactive antibodies

Corresponding Author: Dr Benjamin Meyer

Version 0:

Reviewer comments:

Reviewer #1

(Remarks to the Author)

Bonifaz et al. analyzed serum samples from 2 previously completed clinical studies: (i) samples collected after 1 dose of seasonal influenza vaccine in the 2023/2024 season ("2023 cohort"), and (ii) samples collected after 1 dose of adjuvanted pH1N1 (pH1N1/AS03) influenza vaccine in Nov/Dec 2009 ("2009 cohort"). Their goal was to compare these vaccination strategies for induction of IgG against clade 2.3.4.4b H5N1 viruses currently widespread in birds and dairy cattle. The large size and birth year range of the 2009 cohort allowed the authors to also investigate the effect of early life H1N1 imprinting on the vaccine response. The authors conclude that the pH1N1/AS03 vaccine substantially induced H5N1 cross-reactive antibodies, primarily against the conserved group 1 HA stalk domain. In contrast, the effect was marginal for the unadjuvanted seasonal influenza vaccine. In addition, the authors conclude that pH1N1/AS03 vaccination overcomes the effect of immune imprinting on circulating antibody levels that cross-react with H5. Overall, data are well presented with appropriate statistical analyses.

A major concern is that conclusions are drawn from a comparison of 2 cohorts that differ in composition of the influenza vaccine received and, more importantly, the time (relative to emergence of the 2009 pandemic H1N1 virus) of vaccine administration. The authors acknowledge these limitations, but do not fully consider the implications when drawing their conclusions. The authors note that recipients of the pH1N1/AS03 vaccine would have been immunologically naïve to the pandemic H1 strain. Putting this more precisely, the head domain of the pH1 strain would have been largely novel to most of the vaccine recipients, but all recipients would have had a pool of memory B cells (MBCs) reactive to the H1 stalk because of its high conservation among H1 strains. In the absence of a pool of MBCs with high affinity for the immunodominant HA head to compete (for antigen) with MBCs reactive to the stalk, stalk reactive MBCs would be activated and mediate strong anti-stalk antibody production. This response would be enhanced by adjuvant but would still occur without adjuvant. The situation is very different for individuals in the 2023 cohort. Multiple exposures to the pH1 strain (or highly related variants) through infection and vaccination in these individuals would have generated and expanded high affinity MBCs reactive to the head domain of the H1 in the 2023/2024 seasonal influenza vaccine. As a result, head reactive MBCs would outcompete stalk-reactive MBCs in the response to the vaccine and stalk-reactive antibody production would be minimal (see <https://doi.org/10.1073/pnas.1118979109> and reviews from the groups of AH Ellebedy and PC Wilson). The key point is that conclusions about an adjuvant effect cannot be drawn from a comparison of responses to pH1N1/AS03 vaccination in 2009 and seasonal influenza vaccination in 2023/2024.

An interesting component of the authors analysis focuses on the 2009 cohort to investigate early life imprinting effects on the response to pH1N1/AS03 vaccination. This is an excellent cohort for this analysis because of the large number of subjects and range of birth years. The analysis nicely associates H1N1 imprinting with increased levels of circulating anti-H1 stalk and anti-H5 antibodies at baseline (as recently described by others and referenced in the manuscript – ref 67). A concern is the conclusion that pH1N1/AS03 vaccination "overcomes" the imprinting effect by boosting anti-H1 stalk antibody levels in non-H1N1 imprinted subjects. Antibody levels were only measured one month after vaccination, and it is not known whether the higher levels are maintained. This would presumably require formation of long-lived plasma cells, which generally does not occur in later life. It seems that there is potential to develop the analysis of the response in the 2009 cohort to provide important information. Can the magnitude of the response to pH1N1/AS03 vaccination be determined for individual subjects (perhaps by calculating Delta from day 0-28), and compared for imprinted and non-imprinted subjects? Does the result fit with findings for H5 vaccination in humans?

Other points

1. The pH1N1/AS03 vaccine is described as containing 3.75 µg of H1N1 antigen (line 79). Does the 3.75 µg refer to all viral proteins, or is it a subunit/split virus vaccine that contains 3.75 µg H1?
2. pH1N1/AS03 influenza vaccine is the defined abbreviation of the AS03 adjuvanted pandemic H1N1 influenza vaccine (line 62). It would be helpful to the reader to use this abbreviation throughout, including in figure legends. The vaccine used should be identified in the Figure 5 legend.
3. Antigens with pandemic potential are referred to in the manuscript as having limited immunogenicity (line 357). This was the conclusion from early studies that measured antiviral antibody induction by MN or HAI assays, which measure high affinity anti-head antibodies. The negative results reflected the novelty of the HA head and absence of pre-existing high affinity head-reactive MBCs. A single dose of a pandemic H1 vaccine does however generate antibodies against the stalk, reflecting pre-existing B cell memory for the stalk generated by seasonal HA exposure. Thus, pandemic antigens do not have limited immunogenicity; the response reflects the nature of B cell memory in recipients.
4. Immunodominance of the HA head domain relative to the stalk domain and the competitive dominance of HA head-reactive MBCs relative to stalk-reactive MBCs are key factors determining responses to novel HA vaccination. They should be discussed in the context of study findings.

Reviewer #2

(Remarks to the Author)

Summary

The manuscript reports the H5N1 neutralizing and HA/NA antibody binding serum responses of individuals from two influenza vaccine cohorts, one that received a pH1N1/AS03 vaccine in 2009 and one that received a non-adjuvanted seasonal influenza vaccine in 2023. Using VSV-based pseudovirus, the authors detect H5N1 neutralizing antibodies that significantly correlate with H5 HA-specific, as well as group 1 stalk-specific IgG levels in pre- and post-vaccination sera. Sera pseudo-neutralization additionally correlated with serum inhibition of foci size, or spread, of authentic H5N1 virus. The authors report pH1N1/AS03 vaccination significantly increases H5N1 cross-reactive antibody titers while noting a marginal effect resulting from non-adjuvanted influenza vaccination. They also detect different levels of H5N1 serum neutralization titers pre-vaccination in different aged cohorts in the 2009 vaccine study but equivalent levels 1 month post vaccination. The paper is well-organized and easy to follow but confounding factors in the study design raise questions as to the validity of the conclusions as detailed below.

Significance

H5N1 2.3.4.4b viruses are spreading worldwide, causing sizeable outbreaks in terrestrial and marine animals, with sporadic infections in humans. Detailed characterization of the humoral response against H5N1, for which the human population is largely immunologically naïve, strengthens our understanding of the effects of pre-existing seasonal immunity on the induction of antibody responses towards novel antigens.

Major concerns

1. The authors compare the H5 serum neutralizing titers upon vaccination with an adjuvanted pandemic H1N1 vaccine given in 2009, and therefore first exposure to this antigenically novel H1, with a nonadjuvanted seasonal vaccine given in 2023 and conclude (or at least strongly suggest) that the adjuvant in the 2009 vaccine is responsible for the increase in H5 serum neutralizing titers. However, multiple reports have demonstrated that upon first exposure to a novel influenza strain, the antibody response skews toward broadly reactive stalk-specific responses, and upon subsequent exposures reverts back to the strain-specific HA head directed responses. The authors inappropriately attribute the observed increase in H5N1 cross-neutralizing titer to use of the AS03 adjuvant and fail to sufficiently contextualize the well-published stem-directed antibody response induced by pH1N1 among individuals who are immunologically naïve to pandemic H1N1. While this concern is mentioned in the Discussion, it is more than a limitation of the study, and instead a confounding factor that makes it impossible to draw any conclusions with the regard to the effect of the adjuvant on increasing H5 neutralization titers.
2. The authors also conclude that the pH1N1/AS03 vaccine can overcome immune imprinting patterns of H5N1 cross-reactive titers, but this claim should be nuanced and better explained. While it is true that the peak Ab titers at 1 month post-vaccination are equivalent, despite different pre-vaccination levels, immune imprinting patterns of H5N1 cross-reactive titers may persist as the vaccine response wanes. In addition, no evidence is provided that a non-adjuvanted vaccine couldn't overcome immune imprinting as well.

Overall, while adjuvants may very well be helpful, the design of this study precludes drawing the conclusions the authors make on the effect of an adjuvant in inducing cross-reactive H5N1 antibody responses.

Minor concerns

1. What is the rationale for choosing A/Pelican/Bern/1/2022 and not one of the 2.3.4.4.b H5N1 strains currently circulating, such as A/Texas/24? How different are avian A/Bern/22 and human A/Texas/24?
2. Figure 2 is blurry and it is difficult to read stars indicating p-values. In addition, while correlations are useful it would add to the paper to also include graphs showing the magnitude of binding to the different antigens, not just correlation.

Reviewer #3

(Remarks to the Author)

I co-reviewed this manuscript with one of the reviewers who provided the listed reports. This is part of the Nature

Communications initiative to facilitate training in peer review and to provide appropriate recognition for Early Career Researchers who co-review manuscripts.

Version 1:

Reviewer comments:

Reviewer #1

(Remarks to the Author)

Bonifaz et al. have responded to reviewers' comments and revised and improved their manuscript.

Points to consider

1. In the Abstract, the authors state that pH1N1/AS03 vaccination increases H5N1 cross-reactive antibodies significantly in a pH1N1 immunologically naïve population. This statement is not strictly correct; this population is not completely immunologically naïve to pH1N1. The population may be largely immunologically naïve to the H1 head domain (at least those born after about 1947), but has strong B cell memory to the conserved H1 stalk (the basis of the induction of anti-stalk antibodies by pH1N1/AS03). Individuals also have B cell memory to the N1 (conserved with N1 of seasonal H1N1).
2. The last sentence of the Abstract refers to erasure of imprinted patterns of H5N1 cross-reactive antibody levels by pH1N1/AS03 vaccination. The key basis for this effect is the novelty of the immunodominant H1 head domain in most of the vaccinated recipients in 2009. It is important to be clear that this effect would not be the same if the pH1N1/AS03 vaccine was administered today (see my comments in initial review). This comment also relates to the last sentence of the Introduction.
3. It is puzzling that there is no mention in the Abstract of anti-H1 stalk antibodies. These are of central importance in the manuscript – their correlation with H5N1 cross-neutralization (Fig. 2), their induction by pH1N1/AS03 vaccination (Fig 5; Suppl Fig 1), and their basis for imprinted patterns of H5 cross-reactivity that are erased by pH1N1/AS03 vaccination (Fig 5).
4. In my initial review, I commented on the analysis of birth year-specific differences in H5N1 cross-reactive antibody levels, and elimination of these differences by boosting with pH1N1/AS03 vaccine (Fig 5). I asked whether the magnitude of the response to pH1N1/AS03 vaccine could be determined for individual subjects and compared for H1N1-imprinted and non-H1N1-imprinted subjects. I suggested calculating the delta from day 0 to day 28. By this, I meant the difference between the values for d0 and d28 (d28 MFI minus d0 MFI). This is a better measure of the amount of antibodies produced than is fold-change, which can be very large when the d0 value is low. It appears that the authors have calculated fold-change (not delta), which is shown in Suppl Fig 4. The fold-change analysis is informative and indicates a strong response in non-imprinted cohorts, with little response in the imprinted cohort. This is an important observation. If I have understood what the authors have done, Suppl Fig 4 should be corrected to show that it is a fold-change calculation, not delta. I still recommend calculating delta; it is important for full evaluation of the magnitude of the response and often gives a different picture from fold-change.
5. Related to point 4 above, the added sentence in lines 327-328 is a little confusing and should be rephrased to read something along the lines of: "This largely reflected a significantly stronger antibody response by individuals imprinted with group 2 HA viruses."
6. Figures: Identify the vaccine used in figure legends (Suppl Fig 4) and be consistent in use of vaccine abbreviation (Suppl Fig 5).

Reviewer #2

(Remarks to the Author)

The authors have made an attempt to address reviewer concerns however, the small textual changes do not sufficiently overcome my main concerns.

I still find the major conclusion of the paper to be short of appropriately attributing the increase in H5N1 cross-neutralizing antibodies to the induction of group 1 stalk antibodies following pH1N1/AS03 vaccination in pH1N1 naïve individuals. While the authors have added language to explain the immunological backgrounds of the 2009 and 2023 cohorts and have removed a sentence or two directly comparing adjuvanted and nonadjuvanted vaccines, no statement directly concludes that group 1 HA-specific stalk antibodies induced because the 2009 cohort was pH1N1 naïve are likely responsible for the observed increase in H5N1 cross-neutralizing antibodies. Especially given lines 374-380, the paper still reads in many places as if the adjuvant is responsible for the increase. The fact remains that this paper is evaluating the response to pH1N1/AS03 in a pH1N1 naïve population and only if they were evaluating the response to pH1N1 with or without AS03 in a 2009 cohort, could any conclusions be made. This paper does not demonstrate in any way that if the same vaccine were to be given now in a pH1N1 immune population, it would induce H5N1 cross-neutralizing antibodies. The finding that pH1N1/AS03 is inducing a H5N1 cross-neutralizing response in a pH1N1 naïve population is by itself expected and not novel given the extensive literature demonstrating preferential induction of stalk-specific responses upon first exposure to pH1N1.

In addition, in the discussion, the authors cite a comparison of H5 vaccination with and without AS03 where greater head-specific responses were seen with AS03 (not stalk) and then suggest that the cross-neutralizing H5N1 responses with pH1N1/AS03 vaccine could be directed towards conserved head-directed sites, such as trimer-interface (line 420). However, this is unlikely to be true as trimer interface antibodies are generally non-neutralizing (ref 67 describes a non-neutralizing trimer interface antibody).

The manuscript also continues to state that pH1N1/AS03 “overcomes” or “abolished” immune imprinting. While this vaccine induced a strong H5N1 cross-reactive antibody response that at one month was equivalent in cohorts with differing baseline levels, evaluation of serum titers at 1 month is insufficient to draw such strong conclusions about imprinting and without a contemporary non-adjuvanted cohort, little can be said about the effect of the adjuvant on that response. AS03 has been shown in many contexts to be a good adjuvant and should be considered in contemporary vaccine strategies, but because of the lack of appropriate comparators, no conclusions can be made with the data presented in this manuscript.

Rebuttal letter

Please note that all line numbers indicated below refer to the document with track changes visible.

Editor comments:

Please include the information regarding the clinical reporting that has been provided to the editors prior to the peer-review process within your manuscript. In particular, any previous references to publications of the primary outcomes and details on which pre-specified and/or exploratory analysis you are reporting here.

Answer: We have included the information in the cohort description in the material and methods section (line 98-100) as well as in the reporting summary.

REVIEWER COMMENTS

Reviewer #1 (Remarks to the Author):

Bonifaz et al. analyzed serum samples from 2 previously completed clinical studies: (i) samples collected after 1 dose of seasonal influenza vaccine in the 2023/2024 season (“2023 cohort”), and (ii) samples collected after 1 dose of adjuvanted pH1N1 (pH1N1/AS03) influenza vaccine in Nov/Dec 2009 (“2009 cohort”). Their goal was to compare these vaccination strategies for induction of IgG against clade 2.3.4.4b H5N1 viruses currently widespread in birds and dairy cattle. The large size and birth year range of the 2009 cohort allowed the authors to also investigate the effect of early life H1N1 imprinting on the vaccine response. The authors conclude that the pH1N1/AS03 vaccine substantially induced H5N1 cross-reactive antibodies, primarily against the conserved group 1 HA stalk domain. In contrast, the effect was marginal for the unadjuvanted seasonal influenza vaccine. In addition, the authors conclude that pH1N1/AS03 vaccination overcomes the effect of immune imprinting on circulating antibody levels that cross-react with H5. Overall, data are well presented with appropriate statistical analyses.

A major concern is that conclusions are drawn from a comparison of 2 cohorts that differ in composition of the influenza vaccine received and, more importantly, the time (relative to emergence of the 2009 pandemic H1N1 virus) of vaccine administration. The authors acknowledge these limitations, but do not fully consider the implications when drawing their conclusions. The authors note that recipients of the pH1N1/AS03 vaccine would have been immunologically naïve to the pandemic H1 strain. Putting this more precisely, the head domain of the pH1 strain would have been largely novel to most of the vaccine recipients, but all recipients would have had a pool of memory B cells (MBCs) reactive to

the H1 stalk because of its high conservation among H1 strains. In the absence of a pool of MBCs with high affinity for the immunodominant HA head to compete (for antigen) with MBCs reactive to the stalk, stalk reactive MBCs would be activated and mediate strong anti-stalk antibody production. This response would be enhanced by adjuvant but would still occur without adjuvant. The situation is very different for individuals in the 2023 cohort. Multiple exposures to the pH1 strain (or highly related variants) through infection and vaccination in these individuals would have generated and expanded high affinity MBCs reactive to the head domain of the H1 in the 2023/2024 seasonal influenza vaccine. As a result, head reactive MBCs would outcompete stalk-reactive MBCs in the response to the vaccine and stalk-reactive antibody production would be minimal (see <https://doi.org/10.1073/pnas.1118979109> and reviews from the groups of AH Ellebedy and PC Wilson). The key point is that conclusions about an adjuvant effect cannot be drawn from a comparison of responses to pH1N1/AS03 vaccination in 2009 and seasonal influenza vaccination in 2023/2024.

Answer: We agree with the reviewer that no direct conclusion on the effect of the AS03 adjuvant can be drawn and have therefore acknowledged this fact in the limitations section. However, it seems that we unintentionally gave the impression that we intended to perform a direct comparison of adjuvanted and unadjuvanted influenza vaccines. We have now separated the discussion of adjuvanted and seasonal influenza vaccine and have added a sentence that clearly states why the 2 cohorts cannot be directly compared and adapted our conclusion. In addition, we now discuss HA head and stalk specific antibodies responses in the different cohorts extensively (line 384-428).

An interesting component of the authors analysis focuses on the 2009 cohort to investigate early life imprinting effects on the response to pH1N1/AS03 vaccination. This is an excellent cohort for this analysis because of the large number of subjects and range of birth years. The analysis nicely associates H1N1 imprinting with increased levels of circulating anti-H1 stalk and anti-H5 antibodies at baseline (as recently described by others and referenced in the manuscript – ref 67). A concern is the conclusion that pH1N1/AS03 vaccination “overcomes” the imprinting effect by boosting anti-H1 stalk antibody levels in non-H1N1 imprinted subjects. Antibody levels were only measured one month after vaccination, and it is not known whether the higher levels are maintained. This would presumably require formation of long-lived plasma cells, which generally does not occur in later life. It seems that there is potential to develop the analysis of the response in the 2009 cohort to provide important information. Can the magnitude of the response to pH1N1/AS03 vaccination be determined for individual subjects (perhaps by calculating Delta from day 0-28), and compared for imprinted and non-imprinted subjects? Does the result fit with findings for H5 vaccination in humans?

Answer: We did not determine whether overcoming the effect of immune imprinting is temporary or long-lasting. Therefore, we have added this point to the limitations section (line 476-478). However, we analysed delta of antibody responses before and after vaccination and compared the different birth year groups. Higher increase in antibody titers was observed in group 2 HA compared to group 1 HA imprinted subjects

(supplementary Figure 4). After vaccination with a non-adjuvanted H5N1 vaccines at a high dose of 45ug of HA (12x higher than the HA content of the pH1N1/AS03 vaccine), Garretson et al. found that also higher fold changes in group 2 HA compared to group1 HA imprinted individuals. However, the differences in immune imprinting patterns were only diminished and did not vanish completely. Also, it can be speculated that the markedly increase HA dose used, might have compensated for the lack of an adjuvant in this vaccine formulation. We have added this point to the discussion section. (line 461-467)

Other points

1. The pH1N1/AS03 vaccine is described as containing 3.75 µg of H1N1 antigen (line 79). Does the 3.75 µg refer to all viral proteins, or is it a subunit/split virus vaccine that contains 3.75 µg H1?

Answer: The pH1N1/AS03 vaccine is a subunit/split virus vaccine that contains 3.75ug of H1 HA. We have clarified the section in the manuscript (line 95).

2. pH1N1/AS03 influenza vaccine is the defined abbreviation of the AS03 adjuvanted pandemic H1N1 influenza vaccine (line 62). It would be helpful to the reader to use this abbreviation throughout, including in figure legends. The vaccine used should be identified in the Figure 5 legend.

Answer: We apologize for this oversight and have corrected the abbreviation throughout the text.

3. Antigens with pandemic potential are referred to in the manuscript as having limited immunogenicity (line 357). This was the conclusion from early studies that measured antiviral antibody induction by MN or HAI assays, which measure high affinity anti-head antibodies. The negative results reflected the novelty of the HA head and absence of pre-existing high affinity head-reactive MBCs. A single dose of a pandemic H1 vaccine does however generate antibodies against the stalk, reflecting pre-existing B cell memory for the stalk generated by seasonal HA exposure. Thus, pandemic antigens do not have limited immunogenicity; the response reflects the nature of B cell memory in recipients.

Answer: We thank the reviewer for this comment and have modified the sentence that now states that AS03 induces more broadly reactive antibody responses compared to non-adjuvanted vaccines (line 384)

4. Immunodominance of the HA head domain relative to the stalk domain and the competitive dominance of HA head-reactive MBCs relative to stalk-reactive MBCs are key factors determining responses to novel HA vaccination. They should be discussed in the context of study findings.

Answer: Dominance of HA head over stalk MBCs are now discussed (line 384-428).

Reviewer #2 (Remarks to the Author):

Summary

The manuscript reports the H5N1 neutralizing and HA/NA antibody binding serum responses of individuals from two influenza vaccine cohorts, one that received a pH1N1/AS03 vaccine in 2009 and one that received a non-adjuvanted seasonal influenza vaccine in 2023. Using VSV-based pseudovirus, the authors detect H5N1 neutralizing antibodies that significantly correlate with H5 HA-specific, as well as group 1 stalk-specific IgG levels in pre- and post-vaccination sera. Sera pseudo-neutralization additionally correlated with serum inhibition of foci size, or spread, of authentic H5N1 virus. The authors report pH1N1/AS03 vaccination significantly increases H5N1 cross-reactive antibody titers while noting a marginal effect resulting from non-adjuvanted influenza vaccination. They also detect different levels of H5N1 serum neutralization titers pre-vaccination in different aged cohorts in the 2009 vaccine study but equivalent levels 1 month post vaccination. The paper is well-organized and easy to follow but confounding factors in the study design raise questions as to the validity of the conclusions as detailed below.

Significance

H5N1 2.3.4.4b viruses are spreading worldwide, causing sizeable outbreaks in terrestrial and marine animals, with sporadic infections in humans. Detailed characterization of the humoral response against H5N1, for which the human population is largely immunologically naïve, strengthens our understanding of the effects of pre-existing seasonal immunity on the induction of antibody responses towards novel antigens.

Major concerns

1. The authors compare the H5 serum neutralizing titers upon vaccination with an adjuvanted pandemic H1N1 vaccine given in 2009, and therefore first exposure to this antigenically novel H1, with a nonadjuvanted seasonal vaccine given in 2023 and conclude (or at least strongly suggest) that the adjuvant in the 2009 vaccine is responsible for the increase in H5 serum neutralizing titers. However, multiple reports have demonstrated that upon first exposure to a novel influenza strain, the antibody response skews toward broadly reactive stalk-specific responses, and upon subsequent exposures reverts back to the strain-specific HA head directed responses. The authors inappropriately attribute the observed increase in H5N1 cross-neutralizing titer to use of the AS03 adjuvant and fail to sufficiently contextualize the well-published stem-directed antibody response induced by pH1N1 among individuals who are immunologically naïve to pandemic H1N1. While this concern is mentioned in the Discussion, it is more than a limitation of the study, and instead a confounding factor that makes it impossible to draw any conclusions with the regard to the effect of the adjuvant on increasing H5 neutralization titers.

Answer: We thank the reviewer for this comment. Since reviewer 1 has raised the same concerns, we would like to refer to the answer given to reviewer 1 above.

2. The authors also conclude that the pH1N1/AS03 vaccine can overcome immune imprinting patterns of H5N1 cross-reactive titers, but this claim should be nuanced and

better explained. While it is true that the peak Ab titers at 1 month post-vaccination are equivalent, despite different pre-vaccination levels, immune imprinting patterns of H5N1 cross-reactive titers may persist as the vaccine response wanes. In addition, no evidence is provided that a non-adjuvanted vaccine couldn't overcome immune imprinting as well.

Answer: We agree with the reviewer that we cannot exclude that non-adjuvanted vaccines could not overcome immune imprinting and that we did not investigate antibody titers at a later time point after vaccination. We have added these points to the limitations section (line 476-468).

Overall, while adjuvants may very well be helpful, the design of this study precludes drawing the conclusions the authors make on the effect of an adjuvant in inducing cross-reactive H5N1 antibody responses.

Answer: We agree with the reviewer and have now highlighted this point clearer in the discussion and conclusion.

Minor concerns

1. What is the rationale for choosing A/Pelican/Bern/1/2022 and not one of the 2.3.4.4.b H5N1 strains currently circulating, such as A/Texas/24? How different are avian A/Bern/22 and human A/Texas/24?

Answer: The study was designed and antigens produced before the outbreak of H5N1 clade 2.3.4.4b started in dairy cows in Feb. 2024. Therefore, we had chosen a local strain of H5N1. A/dairy cow/Texas/2024 and A/Pelican/Bern/1/2022 have pairwise sequence identity of 98.8%

2. Figure 2 is blurry and it is difficult to read stars indicating p-values. In addition, while correlations are useful it would add to the paper to also include graphs showing the magnitude of binding to the different antigens, not just correlation.

Answer: We apologize for the blurry graphs and have updated figure 2. We have also included graphs that show the magnitude of binding antibodies as supplementary figure 1 and referenced it in the text accordingly (line 271).

REVIEWER COMMENTS

Reviewer #1 (Remarks to the Author):

Bonifaz et al. have responded to reviewers' comments and revised and improved their manuscript.

Points to consider

1. In the Abstract, the authors state that pH1N1/AS03 vaccination increases H5N1 cross-reactive antibodies significantly in a pH1N1 immunologically naïve population. This statement is not strictly correct; this population is not completely immunologically naïve to pH1N1. The population may be largely immunologically naïve to the H1 head domain (at least those born after about 1947), but has strong B cell memory to the conserved H1 stalk (the basis of the induction of anti-stalk antibodies by pH1N1/AS03). Individuals also have B cell memory to the N1 (conserved with N1 of seasonal H1N1).

Answer: We agree with the reviewer and have changed the phrase to “immunologically partially naïve to pH1N1”.

2. The last sentence of the Abstract refers to erasure of imprinted patterns of H5N1 cross-reactive antibody levels by pH1N1/AS03 vaccination. The key basis for this effect is the novelty of the immunodominant H1 head domain in most of the vaccinated recipients in 2009. It is important to be clear that this effect would not be the same if the pH1N1/AS03 vaccine was administered today (see my comments in initial review). This comment also relates to the last sentence of the Introduction.

Answer: We thank the reviewer for this helpful comment and have deleted this particular statement from the abstract and the title as we felt that the brevity of the abstract is not enough to explain the complex effect of pre-existing immunity on the induction cross-reactive H5N1 antibodies. In line with comments from reviewer 2 we have also adapted the discussion to explain this more in detail and speculate on the added effect of AS03 shown by other studies (line 446-457).

3. It is puzzling that there is no mention in the Abstract of anti-H1 stalk antibodies. These are of central importance in the manuscript – their correlation with H5N1 cross-neutralization (Fig. 2), their induction by pH1N1/AS03 vaccination (Fig 5; Suppl Fig 1), and their basis for imprinted patterns of H5 cross-reactivity that are erased by pH1N1/AS03 vaccination (Fig 5).

Answer: Correlation of H5N1 cross-neutralizing antibodies with group 1 HA stalk and trimeric HA antibodies is now mentioned in the abstract.

4. In my initial review, I commented on the analysis of birth year-specific differences in H5N1 cross-reactive antibody levels, and elimination of these differences by boosting with pH1N1/AS03 vaccine (Fig 5). I asked whether the magnitude of the response to pH1N1/AS03 vaccine could be determined for individual subjects and compared for H1N1-

imprinted and non-H1N1-imprinted subjects. I suggested calculating the delta from day 0 to day 28. By this, I meant the difference between the values for d0 and d28 (d28 MFI minus d0 MFI). This is a better measure of the amount of antibodies produced than is fold-change, which can be very large when the d0 value is low. It appears that the authors have calculated fold-change (not delta), which is shown in Suppl Fig 4. The fold-change analysis is informative and indicates a strong response in non-imprinted cohorts, with little response in the imprinted cohort. This is an important observation. If I have understood what the authors have done, Suppl Fig 4 should be corrected to show that it is a fold-change calculation, not delta. I still recommend calculating delta; it is important for full evaluation of the magnitude of the response and often gives a different picture from fold-change.

Answer: We intended to calculate the delta but have realized that we made a mistake during analysis. We apologize for this oversight and have corrected Suppl. Figure 4. It now shows the delta of log10-transformed antibody titers (VSV-H5N1 neutralization) or log10-transformed MFI values (H5 and H1 trimer and cH6/1).

5. Related to point 4 above, the added sentence in lines 327-328 is a little confusing and should be rephrased to read something along the lines of: "This largely reflected a significantly stronger antibody response by individuals imprinted with group 2 HA viruses."

Answer: We thank the reviewer for this comment. The sentence now reads "This largely reflects a significant stronger increase in antibody levels in individuals imprinted with group 2 HA viruses" (line 312-313)

6. Figures: Identify the vaccine used in figure legends (Suppl Fig 4) and be consistent in use of vaccine abbreviation (Suppl Fig 5).

Answer: Done as requested.

Reviewer #2 (Remarks to the Author):

The authors have made an attempt to address reviewer concerns however, the small textual changes do not sufficiently overcome my main concerns.

I still find the major conclusion of the paper to be short of appropriately attributing the increase in H5N1 cross-neutralizing antibodies to the induction of group 1 stalk antibodies following pH1N1/AS03 vaccination in pH1N1 naïve individuals. While the authors have added language to explain the immunological backgrounds of the 2009 and 2023 cohorts and have removed a sentence or two directly comparing adjuvanted and nonadjuvanted vaccines, no statement directly concludes that group 1 HA-specific stalk antibodies induced because the 2009 cohort was pH1N1 naïve are likely responsible for the observed increase in H5N1 cross-neutralizing antibodies. Especially given lines 374-380, the paper still reads in many places as if the adjuvant is responsible for the increase. The fact remains that this paper is evaluating the response to pH1N1/AS03 in a pH1N1 naïve

population and only if they were evaluating the response to pH1N1 with or without AS03 in a 2009 cohort, could any conclusions be made.

Answer: We agree with the reviewer that we cannot determine the individual contributions of AS03 and pre-existing immunity (presence of HA stalk MBCs and absence of HA head MBCs) to the induction of H5N1 cross-reactive antibodies due to the absence of an appropriate control group and have stated this in line 369-371. In addition, we have now clearly stated that the partially naïve immune status to pH1N1 of the 2009 population likely contributed to the induction of H5N1 cross-reactive antibodies (line 378-380). However, our data shows that HA stalk antibody only partially explains H5N1 cross-neutralization activity, which is in line with other studies. Given these results, we discuss that AS03 may additionally contribute to the induction of H5N1 cross-neutralizing antibodies, but acknowledge that further studies are required to study this (line 380-416).

This paper does not demonstrate in any way that if the same vaccine were to be given now in a pH1N1 immune population, it would induce H5N1 cross-neutralizing antibodies. The finding that pH1N1/AS03 is inducing a H5N1 cross-neutralizing response in a pH1N1 naïve population is by itself expected and not novel given the extensive literature demonstrating preferential induction of stalk-specific responses upon first exposure to pH1N1.

Answer: We agree that we cannot demonstrate the effect of pH1N1/AS03 on the current population and have added a sentence to the limitations section (461-463). We also agree that there is extensive literature that shows an induction of stalk-specific antibodies after first exposure to pH1N1. Nevertheless, induction of cross-neutralizing H5N1 antibodies has not been shown previously and is novel. Also, the fact that cross-neutralizing antibodies do not inhibit entry of authentic H5N1, as would be expected from HA stalk reactive antibodies, but only inhibit its spread has, to our knowledge, not been demonstrated previously, adding to the novelty of our study.

In addition, in the discussion, the authors cite a comparison of H5 vaccination with and without AS03 where greater head-specific responses were seen with AS03 (not stalk) and then suggest that the cross-neutralizing H5N1 responses with pH1N1/AS03 vaccine could be directed towards conserved head-directed sites, such as trimer-interface (line 420). However, this is unlikely to be true as trimer interface antibodies are generally non-neutralizing (ref 67 describes a non-neutralizing trimer interface antibody).

Answer: We agree with the reviewer that the example of the trimer interface antibodies was not well chosen since indeed these antibodies usually do not show neutralizing activity. Nevertheless, there are HA head specific antibodies with cross-neutralizing activity, e.g. F045-092, even across subtypes from different antigenic groups like H1 H3 and H5, indicating that our point is in general still valid. We have adapted the section accordingly and exchanged the reference. (Line 413-416)

The manuscript also continues to state that pH1N1/AS03 “overcomes” or “abolished”

immune imprinting. While this vaccine induced a strong H5N1 cross-reactive antibody response that at one month was equivalent in cohorts with differing baseline levels, evaluation of serum titers at 1 month is insufficient to draw such strong conclusions about imprinting and without a contemporary non-adjuvanted cohort, little can be said about the effect of the adjuvant on that response. AS03 has been shown in many contexts to be a good adjuvant and should be considered in contemporary vaccine strategies, but because of the lack of appropriate comparators, no conclusions can be made with the data presented in this manuscript.

Answer: We have adapted our conclusion and put less emphasis on the fact that AS03 overcomes immune imprinting patterns, also by removing it from the title of the paper. We also acknowledge that differences in pre-existing immunity largely contribute to overcome. In fact, we discuss that immune imprinting was also overcome by a non-adjuvanted H5 HA containing vaccine given to individuals that were likely also immunologically partially naïve to H5N1 (cross-reactive stalk MBCs and no head-specific MBCs). However, we highlight that this vaccine required a much higher dose (12-fold higher) (line 447-457).